

# CA1 pyramidal cells have diverse biophysical properties, affected by development, experience, and aging

Erin C. McKiernan[1] and Diano F. Marrone[2,3]

[1] Departamento de Física, Facultad de Ciencias, Universidad Nacional Autónoma de México, Ciudad de México, México
[2] Department of Psychology, Wilfrid Laurier University, Waterloo, Ontario, Canada
[3] McKnight Brain Institute, University of Arizona, Tucson, AZ, United States of America

## ABSTRACT

Neuron types (e.g., pyramidal cells) within one area of the brain are often considered homogeneous, despite variability in their biophysical properties. Here we review literature demonstrating variability in the electrical activity of CA1 hippocampal pyramidal cells (PCs), including responses to somatic current injection, synaptic stimulation, and spontaneous network-related activity. In addition, we describe how responses of CA1 PCs vary with development, experience, and aging, and some of the underlying ionic currents responsible. Finally, we suggest directions that may be the most impactful in expanding this knowledge, including the use of text and data mining to systematically study cellular heterogeneity in more depth; dynamical systems theory to understand and potentially classify neuron firing patterns; and mathematical modeling to study the interaction between cellular properties and network output. Our goals are to provide a synthesis of the literature for experimentalists studying CA1 PCs, to give theorists an idea of the rich diversity of behaviors models may need to reproduce to accurately represent these cells, and to provide suggestions for future research.

## INTRODUCTION

Understanding the brain and the activity of the approximately 86 billion neurons it contains (*Azevedo et al., 2009*) is a daunting task. To facilitate experimentation and analysis, neuroscientists often group neurons into types (*Armañanzas & Ascoli, 2015*; *Kepecs & Fishell, 2014*; *Wheeler et al., 2015*; *Sharpee, 2014*). However, it is not clear what should be the ideal classification system. Should we group neurons by function, location, morphology, biophysical properties, or some combination? Which, if any, of these groupings are meaningful for predicting a cell's contribution to cognition? Researchers may erroneously assume that cells in one group are relatively homogeneous (*Roeper, 2013*; *Tabansky, Stern & Pfaff, 2015*), but what is the extent and role of biophysical heterogeneity?

Neurons are often classified by their electrical activity (*Contreras, 2004*; *Nowak et al., 2003*). However, neurons in the same brain area or with similar morphology often show different firing patterns (*Faber, Callister & Sah, 2001*; *McCormick et al., 1985*; *Schubert*

Corresponding author
Erin C. McKiernan,
emckiernan@ciencias.unam.mx

*et al., 2006*; *Markram et al., 2004*), while neurons in different areas or with distinct morphology may have similar activity (*Buchhalter & Dichter, 1991*; *Ostapoff, Feng & Morest, 1994*; *Markram et al., 2004*). Moreover, neurons can switch between firing patterns depending on conditions (*Beurrier et al., 1999*; *Connors & Gutnick, 1990*; *Steriade et al., 1998*). In some cases, compensatory changes in connection strengths or regulation of gene expression may counteract variability in firing patterns to maintain normal microcircuit output (*Marder & Goaillard, 2006*; *Turrigiano, 1999*; *Davis, 2006*). In other cases, variability in the biophysical properties of neurons may alter, and even be vital for, network function (*Padmanabhan & Urban, 2010*). The potential functional significance of cellular heterogeneity suggests it deserves more consideration than it has historically received (*Altschuler & Wu, 2010*).

The hippocampus provides an ideal structure to consider functional cellular heterogeneity, given the abundance of data available from this area, as well as its critical role in learning and memory (*Anderson, Grossrubatscher & Frank, 2014*; *Preston & Eichenbaum, 2013*; *Buzsáki & Moser, 2013*; *Yassa et al., 2011*; *Yassa & Stark, 2011*). Hippocampal pyramidal cells (PCs) have been studied extensively using a variety of stimulation protocols and tasks (*Spruston, 2008*; *Disterhoft, Wu & Ohno, 2004*; *Wiener, 1996*). PCs display diverse electrical behaviors, even under seemingly identical experimental conditions, leading some to suggest the existence of distinct subpopulations (*Graves et al., 2012*; *Geiller, Royer & Choi, 2017*; *Moore, Throesch & Murphy, 2011*; *Lee et al., 2014*).

What are these subpopulations? How should experimental scientists classify PC firing behaviors? How should computational scientists decide which firing behaviors a model must reproduce? Are the biophysical properties of PCs relatively stable, or do they vary under different conditions, such as developmental stages, experience, and aging? We synthesize the literature on electrical activity in CA1 PCs in order to provide an overview of the behavioral repertoire of CA1 PCs, both intrinsically and in the context of network activity. We describe results, some conflicting, on how the biophysical properties of CA1 PCs change under different conditions. Finally, we suggest directions for future research, including: text and data mining of the literature to further explore and quantify the extent of cellular heterogeneity within and across neuron types (*Tripathy et al., 2015*; *Wheeler et al., 2015*); use of dynamical systems theory to study and potentially classify firing behaviors (*Izhikevich, 2007*); and use of mathematical modeling to investigate the effects of cellular heterogeneity on network output (*Prinz, Bucher & Marder, 2004*).

## SURVEY METHODOLOGY

While this was not a systematic review, every effort was made to be as comprehensive and unbiased as possible in our searches and descriptions of the literature. Article searches were done in Google Scholar, PubMed, as well as directly through select open access journal websites (e.g., PLOS, PeerJ, Frontiers) when we were searching for openly-licensed figures of CA1 electrical activity that could be reused. We first narrowed down our searches to the cell type of interest by using combinations of the search terms 'CA1', 'hippocampus', and 'pyramidal cells' or 'pyramidal neurons'. We were particularly interested in the electrical

activity of CA1 PCs, so we used terms such as 'electrophysiology', 'firing patterns' or 'spiking patterns', and 'recordings'. We sometimes used the words 'patch clamp' to find articles that specifically used this technique. However, we did not limit our discussion only to patch recordings, as some sharp electrode and multi-unit studies are also included here. We also searched for articles using terms related to specific features of electrical activity, such as 'afterdepolarization', 'afterhyperpolarization', 'bursting', 'delay to first spike' or 'spike latency', 'spike frequency adaptation' or 'accomodation', among others.

We often used several search terms in combination to identify articles. For example, we searched 'firing patterns AND developmental stages', or 'firing patterns AND learning', or 'firing patterns AND aging' to find studies that looked at electrical behavior of CA1 PCs under different conditions. Since the focus of our review was the wide variety in electrical behaviors displayed by CA1 PCs, we used terms like 'diversity', 'heterogeneity', 'subpopulations', and 'variability' to find articles that explicitly discussed or analyzed differences within this cell population. These words were particularly important when searching for articles on subjects such as long-term potentiation or place cell firing, where the literature is vast. An exhaustive discussion of these topics was not feasible herein, nor the focus of this article. Therefore, we relied on combinatorial searches, such as 'long-term potentiation AND variability' or 'place cells AND subpopulations', to find articles that studied CA1 PC differences with respect to these phenomena.

In all searches, we focused only on studies in mammals, the large majority coming from rats and mice, and a much smaller number from other small mammals such as rabbits or guinea pigs. We did not limit our searches to a particular date range.

## RESPONSES TO SOMATIC CURRENT INJECTION

A common experimental protocol to study neuron responses involves injecting pulses of current into the soma to characterize the resulting changes in membrane potential. Because the precise timing and level of stimulation are known, this protocol permits the quantification of temporal aspects of responsiveness such as onset, offset, and adaptation. In addition, small current steps of varying amplitudes allow for the observation of transitions from resting to spiking states.

### Spike latency

In response to current injection, CA1 PCs can begin spiking ∼20 ms or less after stimulus onset (*Aiken, Lampe & Brown, 1995*; *Kim, Wei & Hoffman, 2005*; *Kim et al., 2008*). Short spike latencies in PCs have been shown, though not quantified, in several studies (*Azouz, Jensen & Yaari, 1996*; *Borde, Cazalets & Buno, 1995*; *Gant et al., 2006*; *Gu et al., 2008*; *Malik et al., 2015*; *Moyer et al., 1992*; *Pedarzani et al., 2001*; *Shao et al., 1999*; *Stackman et al., 2002*; *Tombaugh, Rowe & Rose, 2005*). However, other studies have recorded PCs with spiking delays lasting from around 100 milliseconds (ms) to seconds (*Chen & Bazan, 2005*; *Chu et al., 2010*; *Golding et al., 1999*; *Minge & Bähring, 2011*; *Staff et al., 2000*; *Storm, 1988*). In a single study, CA1 PCs were recorded with delays ranging from ∼10 to 110 ms (*Jung & Hoffman, 2009*). In contrast to studies that report no delay, some studies describe long
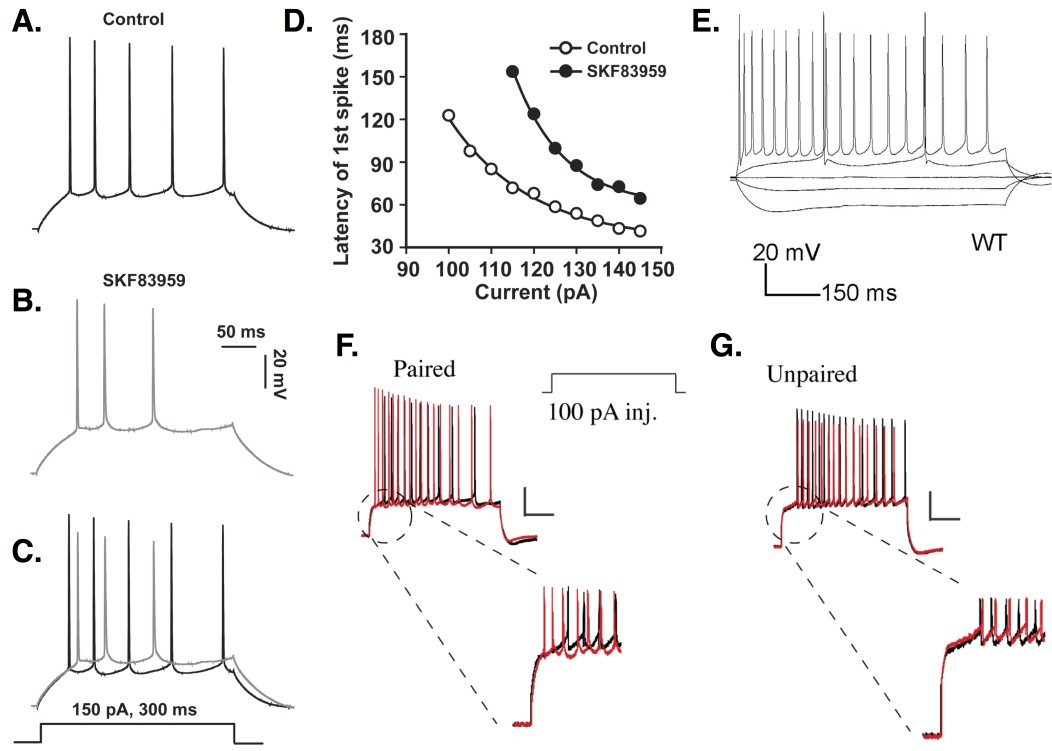

**Figure 1** **Delayed firing in CA1 PCs.** (A–C) An example of moderate spike latency in response to a 150 pA current injection. The delay to first spike is close to 30 ms in control conditions (A), but increases to more than 50 ms in CA1 PCs treated with the dopamine agonist SKF83959 (B); overlap shown in (C). (D) Spike latency ranges from ~30 to over 120 ms, depending on the amplitude of the stimulus current applied, and is increased at all stimulation amplitudes by SKF83959. Images (A–D) are from *Chu et al. (2010)*. (E) Response of CA1 PC to current injections of −150, −50, 0, +50, and +300 pA. Note the response to +50 pA stimulation includes a delay to first spike of over 150 ms. This delay is abolished when the cell is stimulated at +300 pA. Image is from *Ostrovskaya et al. (2014)*. (F) Firing delay is reduced in PCs undergoing a conditioning protocol. (G) Unpaired stimuli do not reduce firing delay. Images (F–G) are from *Jung & Hoffman (2009)*. All figures reused under the terms of the Creative Commons Attribution (CC-BY) license or the Creative Commons Public Domain dedication (CC0).

spike latencies as 'characteristic' (*Golding et al., 1999*) or 'typical' (*Storm, 1988*) of CA1 PC firing. Some of the variability in spike latency is shown in Fig. 1.

Spike latency depends in part on the amplitude of injected current (*Aiken, Lampe & Brown, 1995*; *Chu et al., 2010*; *Golomb, Yue & Yaari, 2006*; *Jensen, Azouz & Yaari, 1994*; *Su et al., 2001*). Aiken et al. report that delay to first spike decreases from 20 to 5 ms as stimulus amplitude increases by 400 pA. Although their baseline delay is longer, *Chu et al. (2010)* similarly report that spike latency decreases by nearly 100 ms as stimulation amplitude increases by 50 pA. Response latency decreases with increasing stimulation amplitude regardless of whether PCs fire single spikes or bursts (*Golomb, Yue & Yaari, 2006*; *Jensen, Azouz & Yaari, 1994*; *Su et al., 2001*).

Spike latency also depends on biophysical properties of PCs, such as availability of $K^+$ channels mediating transmembrane currents. Previous pharmacological studies suggested that slowly-inactivating D-type $K^+$ channels were responsible for producing long firing

delays in CA1 PCs (*Storm, 1988*; *Golding et al., 1999*; *Staff et al., 2000*). Spike latency decreased from hundreds to tens of ms when $I_D$ was blocked (*Storm, 1988*). More recent studies, however, argue that A-type $K^+$ channels, specifically those encoded by the *Kv4.2* gene, play a crucial role in delayed firing (*Varga et al., 2004*; *Kim, Wei & Hoffman, 2005*; *Jung & Hoffman, 2009*). Average delay to first spike observed in PCs increases by hundreds of ms when Kv4.2 currents are potentiated by active CaMKII (*Varga et al., 2004*). Kv4 channel overexpression results in spike latencies tens of ms longer than in controls (*Kim, Wei & Hoffman, 2005*). Loss of the accessory protein DPPX causes the voltage sensitivity of Kv4 channels to shift, leaving more channels available for activation at rest and increasing firing delay in PCs (*Kim et al., 2008*). PCs receiving paired somatic depolarization and synaptic stimulation show decreased spike latency mediated by a reduction in A-type channel availability (*Jung & Hoffman, 2009*) (see Fig. 1).

The hyperpolarization-activated cation current ($I_h$) also affects firing delay in PCs. Application of the $I_h$ blocker ZD7288 causes hyperpolarization of the cell and produces a spike latency of hundreds of ms not seen in controls (*Gu et al., 2005*). In contrast, injecting current to counteract ZD7288-induced hyperpolarization and hold the cell at a normal resting potential eliminates firing delay and causes increased excitability. These results demonstrate the importance of testing various initial conditions when evaluating effects of a given manipulation and the role of different currents. It is likely not one but multiple currents contribute to spike latency, depending on cellular conditions.

The presence or absence of long delays to first spike is not trivial, but rather tells us about the underlying neuronal dynamics. Some cells are not capable of producing long delays at any stimulation amplitude because of the mechanism generating the rest-to-spiking transition, which in turn determines whether these cells are resonators (short delays) or non-resonators (long delays) (*Izhikevich, 2007*). Modeling studies have shown that the type of rest-to-spiking transition observed depends in part on the relative density of $Na^+$ to $K^+$ channels; for example, with the parameters used in *Herrera-Valdez et al. (2013)*, long delays occur when the relative density of these channels is close to one. Experimental studies have shown that CA1 PCs can switch between resonating and non-resonating responses depending on the relative balance of persistent $Na^+$ and M-type $K^+$ currents (*Vera, Alcayaga & Sanhueza, 2017*).

## Spike frequency adaptation

Spike frequency adaptation in response to prolonged current stimulation is often hailed as a characteristic feature of CA1 PC firing. However, the degree of adaptation seen in these cells varies. In some recordings, PCs adapt strongly and cease firing before the end of a current pulse lasting hundreds of ms (*Gant et al., 2006*; *Madison & Nicoll, 1984*; *Stackman et al., 2002*). In other recordings, adaptation slows but does not terminate firing (*Borde, Cazalets & Buno, 1995*; *Gu et al., 2008*; *Jung & Hoffman, 2009*; *Kim, Wei & Hoffman, 2005*; *Malik & Chattarji, 2012*). In some PCs, little to no adaptation is seen (*Chen & Bazan, 2005*; *Malik et al., 2015*; *Springer, Burkett & Schrader, 2015*; *Spruston & Johnston, 1992*; *Staff et al., 2000*).

The degree of adaptation depends in part on stimulus strength (*Bianchi et al., 2012*; *Gu et al., 2008*; *Madison & Nicoll, 1984*). *Bianchi et al. (2012)* report that PCs fire for the duration of a 1 s pulse and spike number increases linearly for moderate stimulation amplitudes (200–900 pA), but decreases at higher amplitudes (over 900 pA). Other studies report only an increase in spike number with no firing cessation even up to 1.3 nA (*Maggio & Segal, 2009*). The change in adaptation at higher stimulus amplitudes seen in some PCs can cause early spikes to cluster into burst-like firing (*Gu et al., 2008*; *Yue et al., 2005*).

Adaptation in PCs shows both a $Ca^{2+}$-dependent component (*Madison & Nicoll, 1984*; *Azouz, Jensen & Yaari, 1996*; *Storm, 1989*) involving L-type channels (*Moyer et al., 1992*), and a non-$Ca^{2+}$-dependent component (*Madison & Nicoll, 1984*). Adaptation may be divided into early and late phases, with distinct currents contributing to each phase (*Gu, Vervaeke & Storm, 2007*; *Pedarzani et al., 2005*; *Storm, 1990*). *Gu et al. (2005)* report a role for M and H channels in early adaptation. Large-conductance $Ca^{2+}$-dependent $K^+$ (BK) channels contribute to early adaptation for high- but not low-frequency firing (*Gu, Vervaeke & Storm, 2007*). There is debate about which currents are responsible for late-phase adaptation. Some studies report a role for small-conductance $Ca^{2+}$-dependent $K^+$ (SK) channels (*Pedarzani et al., 2001*; *Pedarzani et al., 2005*; *Stackman et al., 2002*). *Pedarzani et al. (2001)* found that the SK channel activator 1-EBIO produces such strong adaptation that PCs go from tonic to single spikers. *Gu et al. (2008)* argue, however, that SK channels can slow firing in PCs if necessary, but are not recruited during typical activity. Instead, they show a strong contribution of M channels to adaptation. Other studies confirm that M channels contribute to late adaptation (*Aiken, Lampe & Brown, 1995*), and underlie stronger adaptation in dorsal versus ventral PCs (*Hönigsperger et al., 2015*). It is likely both $Ca^{2+}$-dependent $K^+$ and M currents produce adaptation, depending on conditions (*Madison & Nicoll, 1984*; *Storm, 1989*).

Adaptation in CA1 PCs varies with experience, learning, and aging. Coincident pre- and post-synaptic stimulation decreases adaptation (*Xu et al., 2005*). Firing frequency increases and adaptation decreases in rats exposed to an enriched environment (*Malik & Chattarji, 2012*; *Valero-Aracama, Sauvage & Yoshida, 2015*). Rats trained on an inhibitory avoidance task show decreased adaptation for up to 24 h, while animals exposed to the environment but not trained show a decrease lasting only 1 h (*Farmer & Thompson, 2012*). Similar results are seen in rabbits, where conditioning decreases adaptation 1 h after training (*Moyer Jr, Thompson & Disterhoft, 1996*). Adaptation increases with aging (*Disterhoft et al., 1996*; *Gant et al., 2006*; *Moyer et al., 1992*; *Tombaugh, Rowe & Rose, 2005*) and this increase is associated with impaired learning (*Disterhoft et al., 1996*; *Tombaugh, Rowe & Rose, 2005*).

The presence or absence of adaptation in cells is important not just for experimentalists wanting to characterize responses, but also for modelers to know how to represent these cells. If cells show little to no adaptation, then to reproduce the majority of firing behaviors, it is sufficient to model these cells using a system of only two differential equations where the two variables are voltage and the proportion of activated delayed rectifier $K^+$ channels (*Izhikevich, 2007*; *Av-Ron, Parnas & Segel, 1991*; *Rinzel, 1985*). In contrast, at least three variables are needed to reproduce adaptation (*Izhikevich, 2007*; *Rinzel, 1985*; *Av-Ron, Parnas & Segel, 1993*). Additional variables can include the slow activation of voltage- or

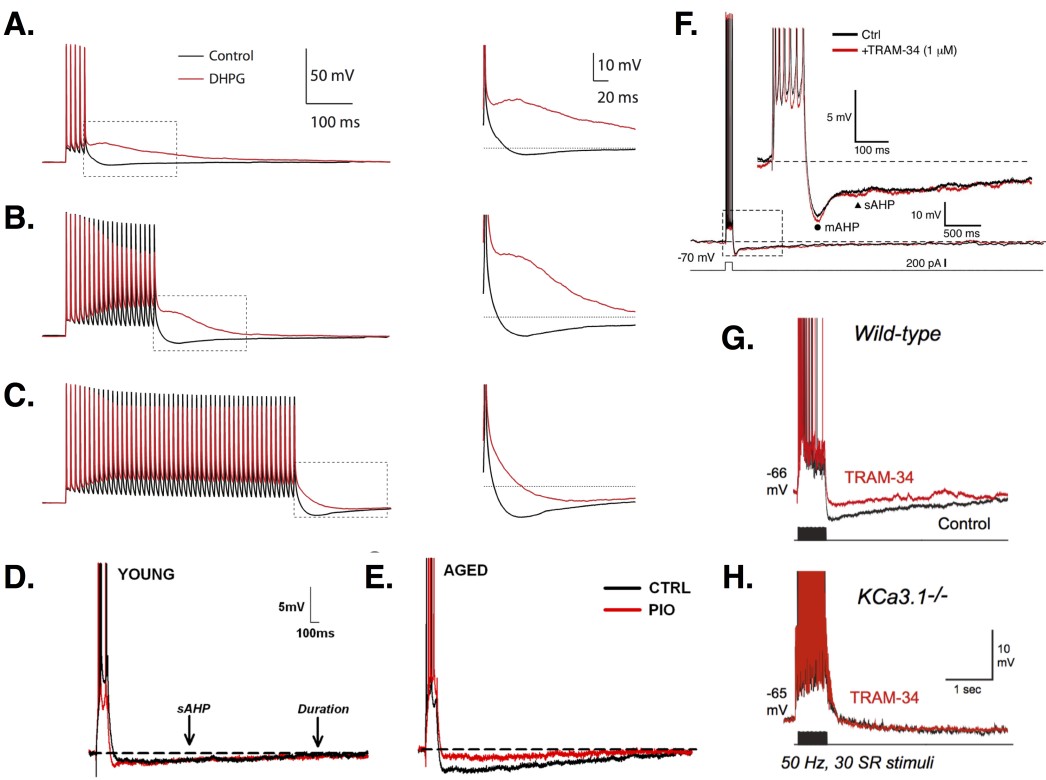

**Figure 2** **AHPs in CA1 PCs.** (A–C) Increasingly long stimulation trains of 5 (A), 20 (B), or 50 (C) APs leads to increased firing and a larger AHP. Treatment with the metabotropic glutamate receptor agonist DHPG (red traces) abolishes the AHP and produces an ADP, except for the longest stimulation train, where the AHP is simply reduced (see zooms). Images (A–C) are from *Park et al. (2010)*. (D–E) Young PCs (E) show smaller AHPs than aged cells (F). In aged but not young PCs, the AHP is reduced by the diabetes drug pioglitazone (PIO; red traces). Images (D–E) are from *Blalock et al. (2010)*. (F) Recordings show the time course of the mAHP and sAHP. In this study, no effect of the intermediate $Ca^{2+}$-dependent $K^+$ channel blocker TRAM-34 was found, suggesting no role for these channels in generating AHPs. Image is from *Wang et al. (2016)*. (G) In contrast, *King et al. (2015)* found that TRAM-34 did reduce the AHP in wild-type CA1 PCs. (H) In KCa3.1 null mice, the sAHP is smaller than in wild-type and TRAM-34 has no additional effect, suggesting a role for these channels in mediating AHPs. Images (G–H) are from *King et al. (2015)*. All figures reused under the CC-BY license or CC0 dedication.

$Ca^{2+}$-gated outward (e.g., $K^+$) currents, or the inactivation of voltage- or $Ca^{2+}$-gated inward (e.g., $Na^+$or $Ca^{2+}$) currents. In other words, there are multiple solutions for creating a 3+-dimensional model that has the capacity for adaptation (*Izhikevich, 2007*).

## Afterhyperpolarizations

Action potential (AP) repolarization and spike frequency adaptation in PCs are mediated by afterhyperpolarizations (AHPs) (see Fig. 2), often divided into fast (1–5 ms), medium (tens to hundreds of ms), and slow (hundreds of ms to seconds) components (*Storm, 1989*; *Storm, 1990*; *Gu et al., 2005*). AHPs are distinguished not only by their timescales, but also by their underlying currents and effects on PC firing.

### fAHPs

The fAHP is mediated by a $Ca^{2+}$-dependent BK current. $Ca^{2+}$-free medium, $Ca^{2+}$ chelators, and $Ca^{2+}$ channel blockers reduce or eliminate the fAHP (*Shao et al., 1999*; *Storm, 1989*; *Su et al., 2001*; *Yue & Yaari, 2004*). Application of BK channel blockers reduces the fAHP, slows AP repolarization, and produces prolonged spikes in PCs (*Shao et al., 1999*; *Springer, Burkett & Schrader, 2015*; *Storm, 1987*; *Gu, Vervaeke & Storm, 2007*). During high-frequency (e.g., 100 Hz) firing, spike broadening between the first and third spikes of a train in response to current injection is eliminated by BK channel blockers when all spikes become broadened (*Shao et al., 1999*). This slows high-frequency repetitive firing and reduces early adaptation (*Gu, Vervaeke & Storm, 2007*). BK channels play a greater role in the early rather than late phase of the response due to channel inactivation (*Shao et al., 1999*; *Gu, Vervaeke & Storm, 2007*). Interestingly, low-frequency (e.g., 13 Hz) firing does not produce spike broadening and is unaffected by BK channel block (*Shao et al., 1999*; *Gu, Vervaeke & Storm, 2007*). Spike prolongation increases as firing frequency goes from 10 to 150 Hz, indicating the increasing influence of BK currents (*Shao et al., 1999*). Repeated synaptic stimulation decreases the somatically-induced fAHP (*Springer, Burkett & Schrader, 2015*). In both young and aged rats, learning decreases fAHP amplitude relative to controls (*Matthews, Linardakis & Disterhoft, 2009*). There is no difference between fAHPs recorded in PCs from young versus aged animals.

### mAHPs

The currents producing the mAHP in CA1 PCs are under debate. Some studies implicate SK channels (*Alger, Sim & Brown, 1994*; *Kramár et al., 2004*; *Stocker, Krause & Pedarzani, 1999*; *Pedarzani et al., 2001*; *Stackman et al., 2002*). Outward currents underlying the mAHP show $Ca^{2+}$ dependence (*Alger, Sim & Brown, 1994*; *Stocker, Krause & Pedarzani, 1999*; *Shah & Haylett, 2000*), and a single-channel conductance of 17–31 pS (*Alger, Sim & Brown, 1994*), similar to SK channels in other systems (*Sah & Faber, 2002*). SK channel blockers reduce the mAHP (*Kramár et al., 2004*; *Stocker, Krause & Pedarzani, 1999*; *Shah & Haylett, 2000*; *Pedarzani et al., 2001*), causing increased PC excitability and improved plasticity and spatial memory (*Stackman et al., 2002*). SK agonists augment the mAHP and reduce excitability, converting some PCs from tonic to single spikers (*Pedarzani et al., 2001*). Studies in transgenic mice have identified SK2-encoded channels as mediating the mAHP (*Bond et al., 2004*).

Other studies suggest SK channels do not contribute to the mAHP in CA1 PCs. *Gu et al. (2005)* and *Gu et al. (2008)* report that only when other channels are blocked is a SK-related portion of the mAHP revealed. Under control conditions and at normal resting potential, the mAHP is reduced by the M channel blocker XE991 (*Gu et al., 2005*; *Gu et al., 2008*), which increases firing frequency and converts PCs from tonic spikers to bursters (*Gu et al., 2005*). At hyperpolarized membrane potentials, the mAHP is reduced by the H channel blocker ZD7288. Thus, $I_M$ and $I_h$ may underlie mAHPs, depending on cellular conditions (*Gu et al., 2005*). Additional studies confirm $I_M$ contributes to the mAHP in CA1 PCs (*Storm, 1989*; *Hönigsperger et al., 2015*). Dorsal PCs are more sensitive than ventral PCs to M current blockers, including a larger reduction in the mAHP current (*Hönigsperger et al.,*

*2015*). *Chen, Benninger & Yaari (2014)* suggest that SK and M channels serve supporting roles, as seen in other hippocampal cells (*Mateos-Aparicio, Murphy & Storm, 2014*). A modeling approach could be helpful here by allowing us to precisely control the balance of SK and M currents and determine under what parameter regimes one current could compensate for the other.

Effects of experience and aging on mAHPs vary. Some studies show age-related increases in the mAHP, beginning in rats at 12 months of age and continuing to increase at 23 months (*Gant et al., 2006*). Other studies report no age differences in the mAHP (*Tombaugh, Rowe & Rose, 2005*). For some tasks, learning in young animals decreases mAHP amplitude, while middle-aged animals classified as learning-impaired show no change (*Kaczorowski & Disterhoft, 2009*). In other tasks, the mAHP does not differ with age, irrespective of learning outcome (*Tombaugh, Rowe & Rose, 2005*).

### sAHPs

The sAHP is $Ca^{2+}$-dependent (*Azouz, Jensen & Yaari, 1996*; *Chen, Yue & Yaari, 2005*; *Madison & Nicoll, 1984*; *Shah & Haylett, 2000*; *Storm, 1989*; *Wu, Chan & Disterhoft, 2004*), and reduced by L-type $Ca^{2+}$ channel blockers (*Moyer et al., 1992*; *Power et al., 2002*; *Shah & Haylett, 2000*; *Su et al., 2001*) and interference with intracellular stores (*Bodhinathan, Kumar & Foster, 2010*; *Gant et al., 2006*; *Shah & Haylett, 2000*). L-type $Ca^{2+}$ channels may be coupled to ryanodine receptors on the stores (*Chavis et al., 1996*; *Gant et al., 2006*). The sAHP is likely mediated by $Ca^{2+}$-dependent $K^+$ channels, though its molecular identity is still under investigation (*Andrade, Foehring & Tzingounis, 2012*). sAHP channel conductance is small (*Andrade, Foehring & Tzingounis, 2012*), ruling out BK channels. SK channels are unlikely to be involved either, since the sAHP is unaffected by SK-specific blockers (*Gu et al., 2005*; *Storm, 1989*; *Stocker, Krause & Pedarzani, 1999*; *Shah & Haylett, 2000*). Transgenic studies in mice have ruled out a role for SK1-3 (*Bond et al., 2004*), and suggest the sAHP is mediated by intermediate-conductance $Ca^{2+}$-dependent KCa3.1 channels (*King et al., 2015*). Recent pharmacological studies using the $I_K$ blocker, TRAM-34, have had mixed results, with some not seeing an effect on the sAHP (*Wang et al., 2016*), while others see a significant reduction (*Turner et al., 2016*; *King et al., 2015*). These differences may be due to drug delivery method (*Turner et al., 2016*).

Experience, learning, and aging affect sAHPs. Repeated square-pulse stimulation increases sAHP amplitude and duration, producing decreased excitability over time (*Borde, Cazalets & Buno, 1995*). Aged animals show larger sAHPs than young animals (*Bodhinathan, Kumar & Foster, 2010*; *Campbell et al., 1996*; *Disterhoft et al., 1996*; *Gant et al., 2006*; *Gant & Thibault, 2009*; *Kaczorowski & Disterhoft, 2009*; *Matthews, Linardakis & Disterhoft, 2009*; *Power et al., 2002*). Environmental enrichment or exercise decreases sAHP amplitude (*Kumar & Foster, 2007*; *Malik & Chattarji, 2012*), abolishing the difference between aged and young animals (*Kumar & Foster, 2007*). Learning also seems to reduce sAHPs, with the duration and latency of reduction depending on the task and the location of the cells. In ventral CA1 PCs in young rats, exposure to an environment reduces sAHPs for up to 1 h, whereas inhibitory training leads to a 24-hour reduction (*Farmer & Thompson, 2012*). Dorsal CA1 PCs show a longer effect latency, with sAHP reduction only after 24 h.

Eyeblink and fear conditioning protocols also decrease sAHPs, but only in successful learners (*Disterhoft et al., 1996*; *Kaczorowski & Disterhoft, 2009*; *Matthews, Linardakis & Disterhoft, 2009*). Aged animals with impaired learning show similar sAHP amplitudes to untrained controls.

## Afterdepolarizations

In response to brief current pulses ($\leq 5$ ms), most CA1 PCs fire a single AP, followed by an afterdepolarization (ADP) (*Azouz, Jensen & Yaari, 1996*; *Jensen, Azouz & Yaari, 1996*; *Kirson & Yaari, 2000*; *Su et al., 2001*; *Yue et al., 2005*). *Jensen, Azouz & Yaari (1994)* and *Jensen, Azouz & Yaari (1996)* report 35–50% of CA1 PCs show "passive" ADPs lasting $\sim 20$ ms and characterized by a smooth decay and a time constant of $\sim 12$ ms. The remaining PCs show "active" ADPs lasting $\sim 40$ ms, with a period of renewed depolarization before decaying, and a time constant of $\sim 20$ ms. Mice from different genetic backgrounds have different-sized ADPs (*Routh et al., 2009*). Some PCs show no obvious ADP (*Zawar et al., 1999*).

ADPs regulate firing patterns in CA1 PCs. Bursting PCs are more likely to have active and larger ADPs compared to non-bursters (*Jensen, Azouz & Yaari, 1994*; *Jensen, Azouz & Yaari, 1996*; *Su et al., 2001*). PCs that burst early rather than late in response to current injection also have larger ADPs (*Graves et al., 2012*). Lowering extracellular $Ca^{2+}$ increases ADP amplitude, changing spiking cells into bursters (*Azouz, Jensen & Yaari, 1996*). In rats, ADP duration increases from $\sim 5$ ms before postnatal day 10 (P10) to a maximum of $\sim 30$ ms at P18, then decreases to $\sim 20$ ms as animals progress to adulthood, but with large variability across PCs. Changes in ADP duration during development are associated with propensity of burst firing (*Chen, Yue & Yaari, 2005*).

Evidence suggests that both $Ca^{2+}$ and persistent $Na^+$ channels participate in ADP generation in CA1 PCs, though the location of each channel population and their importance during developmental stages is different. In both developing (P8-P25) and adult ($P > 30$) rats (*Chen, Yue & Yaari, 2005*), persistent $Na^+$ currents participate in ADP generation. Blocking persistent $Na^+$ channels with drugs applied to the soma but not the dendrites reduces the ADP (*Chen, Yue & Yaari, 2005*; *Yue et al., 2005*). In developing PCs, blocking T/R- and L-type $Ca^{2+}$ channels with drugs applied to the dendrites but not the soma also decreases the ADP and reduces bursting (*Chen, Yue & Yaari, 2005*), while these blockers have little to no effect on ADPs in adult PCs (*Yue et al., 2005*). N/P/Q-type $Ca^{2+}$ channel blockers have no effect on the ADP in either developing or adult cells (*Chen, Yue & Yaari, 2005*). $K^+$ currents active during AP repolarization help control ADP size. Increasing extracellular $K^+$ augments the ADP and can trigger bursting (*Jensen, Azouz & Yaari, 1994*). Blocking M channels at the soma but not the dendrites augments the ADP, whereas SK channel blockers have no effect (*Yue & Yaari, 2004*; *Yue & Yaari, 2006*). In contrast, blocking A-type $K^+$ channels at the dendrites but not the soma increases the ADP and leads to bursting (*Yue & Yaari, 2006*). This effect of A-type channel blockers is thought to be caused by an increase in back-propagating action potentials (bAPs) (*Magee & Carruth, 1999*; *Yue et al., 2005*; *Hoffman et al., 1997*)—APs that travel from the site of initiation near the soma 'backwards' into the dendrites (for reviews see *Magee et al., 1998*; *Waters, Schaefer & Sakmann, 2005*). A-type $K^+$ currents in the dendrites help control the size of bAPs and

lessen or prevent propagation. When A-type currents are blocked by a drug or inactivated by repetitive firing, bAPs can propagate and depolarize the dendrites, sometimes enough to cause an influx of $Ca^{2+}$ that produces a dendritic $Ca^{2+}$ spike. These $Ca^{2+}$ spikes can then propagate to the soma, contributing to an increase in the ADP amplitude and causing bursting (*Magee & Carruth, 1999*; *Yue et al., 2005*). Thus, by regulating the level of A-type channel activity, for example, via protein kinase activity (*Johnston et al., 1999*), cells can control their excitability and potentially switch back and forth between different firing patterns (*Magee & Carruth, 1999*).

## Burst firing

PCs can burst, rather than simply fire repetitively (*Azouz, Jensen & Yaari, 1996*; *Chen, Yue & Yaari, 2005*; *Golomb, Yue & Yaari, 2006*; *Jensen, Azouz & Yaari, 1994*; *Su et al., 2001*; *Sánchez-Aguilera et al., 2017*; *Graves et al., 2012*). Some studies claim PCs can be grouped into three classes based on firing patterns (*Jensen, Azouz & Yaari, 1994*; *Su et al., 2001*). Around 80% of PCs are 'regular spiking cells' or 'non-bursters'. The remaining ∼20% are either (1) 'borderline'/'high-threshold' bursters, which fire single spikes at low stimulation amplitudes but burst at higher amplitudes, or (2) 'low threshold' bursters, which burst at low and high stimulation amplitudes. The latter class can be further divided into three grades. Grade I PCs fire single spikes in response to short stimuli but burst with long (hundreds of ms) pulses. Grade II PCs burst in response to both short and long pulses. Grade III PCs burst in the absence of stimulation (*Jensen, Azouz & Yaari, 1994*; *Su et al., 2001*). Other studies claim adult PCs should be divided into only two classes, those that burst early versus those that burst late in response to current injection (*Graves et al., 2012*). Furthermore, these two bursting cell types show other differences, including distinct dendritic morphologies, arguing these are ''stable cell types'' and not simply the same PCs transitioning between different firing patterns (*Graves et al., 2012*).

The percentage of bursting PCs depends in part on anatomical location. Only ∼10% of proximal (i.e., near CA2/CA1 border) PCs burst, compared to over 50% of distal (i.e., near the subiculum) PCs (*Jarsky et al., 2008*). This difference in bursting likelihood correlates with a similar anatomical variation in ADP amplitude. Changes in the driving forces for $Ca^{2+}$ entry (*Azouz, Jensen & Yaari, 1996*; *Golomb, Yue & Yaari, 2006*; *Su et al., 2001*) and $K^{+}$ exit (*Jensen, Azouz & Yaari, 1994*) from the cell increase ADPs and promote bursting. In PCs from adult rats, reducing extracellular $Ca^{2+}$ can induce bursting (*Golomb, Yue & Yaari, 2006*; *Su et al., 2001*), convert PCs from high- to low-threshold bursters (*Su et al., 2001*), and increase bursting frequency (*Azouz, Jensen & Yaari, 1996*; *Golomb, Yue & Yaari, 2006*). Increasing extracellular $K^{+}$ increases the percentage of bursters and can induce spontaneous, rhythmic bursting (*Jensen, Azouz & Yaari, 1994*).

The percentage of bursting PCs also depends on developmental stage. In rats, nearly all PCs are non-bursters up to postnatal day 11 (P11) (*Chen, Yue & Yaari, 2005*). Subsequently, the percentage of bursters increases, hitting a maximum of 74% between P18 and P19. The percentage decreases again as animals approach adulthood. More recent studies have reported a slightly different but similar timeline, with the majority of cells being non-bursters up to P10, increasing to 100% bursters at P16, and then decreasing again to

around 88% bursters between P18 and P19 (*Sánchez-Aguilera et al., 2017*). These changes in bursting tendency correlate with developmental changes in ADP duration (*Chen, Yue & Yaari, 2005*) and amplitude (*Sánchez-Aguilera et al., 2017*). Studies suggest that the primary mechanism underlying bursting may change during maturation (*Chen, Yue & Yaari, 2005*; *Golomb, Yue & Yaari, 2006*). In developing PCs, bursting depends on bAPs, which activate dendritic $Ca^{2+}$ spikes that then propagate to the soma to boost ADPs and induce bursting (*Chen, Yue & Yaari, 2005*). This mechanism is sometimes referred to as 'ping-pong' bursting (*Chen, Yue & Yaari, 2005*; *Golomb, Yue & Yaari, 2006*) because it involves the activity of channel populations in two different cellular compartments—the soma and the dendrites—and the propagation of signals back and forth between the two. The dependence of young cells on dendritic $Ca^{2+}$ currents means that, in contrast to adult PCs (*Azouz, Jensen & Yaari, 1996*; *Golomb, Yue & Yaari, 2006*), decreasing extracellular $Ca^{2+}$ in young PCs reduces bursting (*Sánchez-Aguilera et al., 2017*). In adult PCs, bursting instead depends almost exclusively on somatic persistent $Na^+$ currents, and little, if at all, on dendritic $Ca^{2+}$ currents (*Yue et al., 2005*). Bursting in adult PCs can occur even when the apical dendrites are truncated (*Yue et al., 2005*; *Golomb, Yue & Yaari, 2006*), showing that only the somatic currents are necessary and suggesting a 'square-wave' bursting mechanism (*Golomb, Yue & Yaari, 2006*). This shift in predominant bursting mechanisms is thought to be due to changes in the relative density of outward and inward currents during developmental (*Chen, Yue & Yaari, 2005*; *Sánchez-Aguilera et al., 2017*), but adult cells can display both types of bursting under the right conditions. When dendritic A-type $K^+$ channels are functional and prohibit bAPs, adult PCs depend more on the square-wave mechanism, but ping-pong bursting can be induced by blocking A-type $K^+$ channels, which allows bAPs to invade the dendrites (*Golomb, Yue & Yaari, 2006*; *Magee & Carruth, 1999*).

As with adaptation, bursting requires at least three variables (*Izhikevich, 2007*; *Av-Ron, Parnas & Segel, 1993*; *Toporikova et al., 2008*; *Rinzel & Ermentrout, 1998*), thus indicating the minimal dimensionality required to accurately reproduce the diversity of firing patterns seen in CA1 PCs, especially if we want to represent various developmental stages. Such a model can be constructed in a variety of ways (*Izhikevich, 2007*), but commonly involves $Ca^{2+}$ dynamics as a third variable (*Av-Ron, Parnas & Segel, 1993*; *Rinzel & Ermentrout, 1998*). If we only want to investigate the role of somatic currents, then a single-compartment model will suffice, while at least two compartments are needed if we want to explore the role of bAPs and dendritic currents in 'ping-pong' bursting (*Golomb, Yue & Yaari, 2006*).

## RESPONSES TO SYNAPTIC STIMULATION

CA1 PCs receive input from neurons in the CA3 region of the hippocampus via the Schaffer collaterals (SC) terminating in the stratum radiatum, as well as directly from the entorhinal cortex. Stimulation of the perforant path (PP) will excite PCs both indirectly via CA3, as well as the direct temporoammonic pathway terminating in the stratum lacunosum-moleculare (*Szilagyi et al., 2011*). Studies have explored SC and PP stimulation to probe responsiveness of CA1 PCs.

## Synaptically-induced afterhyperpolarizations

Synaptic AHPs (synAHPs) are evoked by microsecond pulses delivered to the PP or SCs (*Kramár et al., 2004*; *Lancaster et al., 2001*; *Wu, Chan & Disterhoft, 2004*; *Gant & Thibault, 2009*; *Newberry & Nicoll, 1984*; *Otmakhova & Lisman, 2004*). synAHP size and duration depend on amplitude, location, number, and frequency of the stimulation (*Gant & Thibault, 2009*; *Otmakhova & Lisman, 2004*; *Newberry & Nicoll, 1984*; *Wu, Chan & Disterhoft, 2004*). Burst firing produces larger synAHPs than single spikes (*Gant & Thibault, 2009*). Suprathreshold stimulation eliciting spikes evokes longer-duration synAHPs than subthreshold stimulation eliciting only EPSPs (*Newberry & Nicoll, 1984*; *Wu, Chan & Disterhoft, 2004*). synAHPs following single EPSPs are larger when evoked by PP stimulation versus SC stimulation (*Otmakhova & Lisman, 2004*). Around 20% of PCs show no synAHPs and instead show large post-burst ADPs and prolonged spiking, effectively dividing CA1 PCs into two subpopulations based on this response (*Wu, Chan & Disterhoft, 2004*).

Medium-duration synAHPs in response to subthreshold stimulation are not $Ca^{2+}$-dependent (*Otmakhova & Lisman, 2004*; *Wu, Chan & Disterhoft, 2004*). Instead, these synAHPs are mediated by $I_H$, but to varying extents depending on the source of stimulation (*Otmakhova & Lisman, 2004*). Those elicited by SC stimulation are nearly abolished by the $I_H$ blocker ZD7288, while those produced by PP stimulation are only reduced by half and depend additionally on GABAergic signaling (*Otmakhova & Lisman, 2004*). Like somatically-generated sAHPs, slow synAHPs following bursts are $Ca^{2+}$-dependent (*Kramár et al., 2004*; *Lancaster et al., 2001*). The molecular identity of the underlying channels is unclear. Some studies suggest a role for SK channels (*Kramár et al., 2004*), while others argue against SK involvement based on synAHP kinetics and sensitivity to noradrenaline and $\beta$-adrenergic agonists (*Lancaster et al., 2001*). Still others have recorded synaptically-stimulated hyperpolarizations which do not respond to $I_{KCa}$ antagonists, but are decreased by enkephalin, suggesting a role for extrinsic input from hippocampal interneurons in generating these potentials (*Newberry & Nicoll, 1984*).

Aging has different effects on slow AHPs stimulated somatically versus synaptically. As discussed in 'Afterhyperpolarizations', slow AHPs elicited by somatic current injection are larger in aged versus young adult rats, and contribute to spike failure during repetitive stimulation. In contrast, synaptically-generated AHPs do not affect repetitive spiking, and are smaller in aged than in young adult animals (*Gant & Thibault, 2009*).

## Short-term potentiation

Microsecond-pulse trains delivered to the SCs at 1-15 Hz increase EPSP amplitudes and population spikes in CA1 PCs (*Applegate & Landfield, 1988*; *Landfield, Pitler & Applegate, 1986*; *Landfield, McGaugh & Lynch, 1978*; *Landfield & Morgan, 1984*; *Ouanounou et al., 1999*; *Gant & Thibault, 2009*; *Thibault, Hadley & Landfield, 2001*). This short-term plasticity, called frequency facilitation (FF) or potentiation (FP), occurs in response to both sub- and suprathreshold stimulation (*Thibault, Hadley & Landfield, 2001*). Potentiation ranges from 20–300% across animals (*Landfield & Lynch, 1977*), and varies with anatomical location (*Papatheodoropoulos & Kostopoulos, 2000b*). Dorsal PCs show FP in response to

SC stimulation at 1–40 Hz, with the largest potentiation at 10–20 Hz. Higher frequencies result in depressed EPSPs. Ventral PCs instead show minimal FP at 1 Hz but either no response or depression at higher frequencies tested up to 100 Hz (*Papatheodoropoulos & Kostopoulos, 2000b*).

While subthreshold stimulation produces equivalent FP in young and aged animals (*Thibault, Hadley & Landfield, 2001*), the response to suprathreshold stimulation is reduced with aging (*Thibault, Hadley & Landfield, 2001*; *Applegate & Landfield, 1988*; *Landfield, Pitler & Applegate, 1986*; *Landfield, McGaugh & Lynch, 1978*; *Landfield & Morgan, 1984*; *Ouanounou et al., 1999*). *Gant & Thibault (2009)* report that FP responses are significantly different between aged and young animals at stimulation frequencies in the theta range (e.g., 7 Hz), but not at lower (3 Hz) or higher (15 Hz) frequencies. Other studies, however, report robust FP differences in young versus aged animals at stimulation frequencies of 10–12 Hz (*Landfield, McGaugh & Lynch, 1978*; *Applegate & Landfield, 1988*). *Landfield, McGaugh & Lynch (1978)* report that young animals show a multiphase response to stimulation, with early potentiation, followed by depression, and then stronger potentiation. In contrast, FP in aged animals is normal for the first few pulses, but decays and responses depress with no rebound as stimulation proceeds (*Gant & Thibault, 2009*). Response depression in aged animals is stronger and faster with higher frequencies and longer pulse trains (*Applegate & Landfield, 1988*; *Landfield, McGaugh & Lynch, 1978*).

FP in young animals is associated with a decrease in the distal vesicle pool, an increase in the local pool, and a clustering of vesicles at the synapse active zone (*Applegate & Landfield, 1988*). Aged animals show a lower density of distal vesicles at rest relative to young animals, and show little change in density with stimulation. In addition, while the local vesicle pool increases in aged animals with stimulation, fewer vesicles are found clustered at the active zone, indicating aged animals may have impaired FP as a result of deficits in vesicle release or cycling (*Applegate & Landfield, 1988*).

Development of FP in young animals is associated with increases in intracellular $Ca^{2+}$. Aged animals demonstrate similar increases in $Ca^{2+}$ during the early phase but larger increases during the late phase of repetitive stimulation, compared to young animals (*Thibault, Hadley & Landfield, 2001*). This excess $Ca^{2+}$ is thought to activate $Ca^{2+}$-dependent channels underlying AHPs, which are larger in aged animals and contribute to decreased excitability (*Gant & Thibault, 2009*; *Thibault, Hadley & Landfield, 2001*). Larger AHPs in aged animals can lead to spike failure in response to subsequent synaptic stimulation (*Gant & Thibault, 2009*). Aged rats exposed to high $Mg^{2+}$ show increased FP and better learning (*Landfield & Morgan, 1984*).

### Long-term potentiation

One of the most well-studied forms of plasticity in CA1 PCs is long-term potentiation (LTP) (*Lauri et al., 2007*; *Gruart & Delgado-García, 2007*; *El-Gaby, Shipton & Paulsen, 2014*; *Malenka & Bear, 2004*; *Hölscher, 1999*; *Volianskis, Collingridge & Jensen, 2013*), an increase in synaptic strength in response to repetitive stimulation (see Fig. 3) first reported in the dentate (*Bliss & Lømo, 1973*). A comprehensive discussion of LTP is beyond the

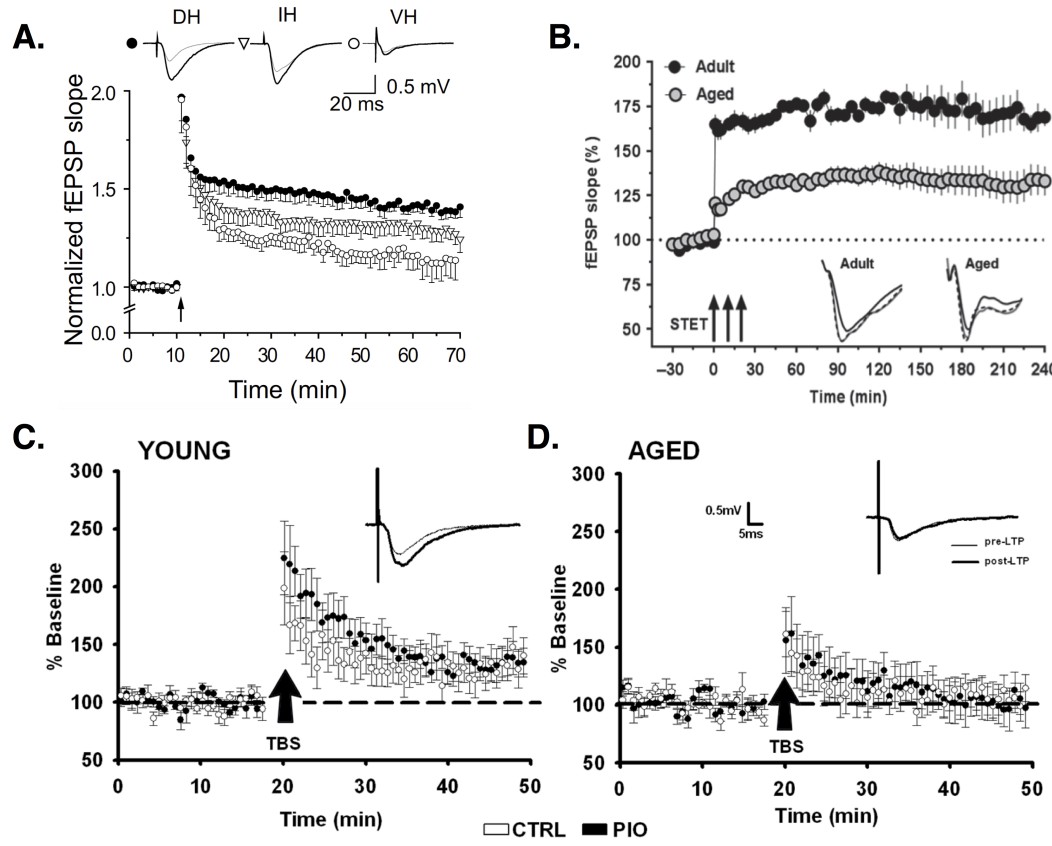

**Figure 3** **Synaptic plasticity in CA1 PCs.** (A) LTP induced by high-frequency stimulation (HFS) is larger in dorsal (filled circles) versus intermediate (open triangles) or ventral (open circles) horn PCs. Image is from *Milior et al. (2016)*. (B) LTP induced by three trains of HFS (arrows; AKA strong tetanization protocol, STET) is reduced in aged (gray circles) versus young adult rats (black circles). Image is from *Shetty, Sharma & Sajikumar (2017)*. (C–D) Baseline EPSPs are larger in young (C) versus aged (D) animals. Potentiation of synaptic responses is reduced or non-existent in aged animals (D). Images (C–D) are from *Blalock et al. (2010)*. All figures reused under the CC-BY license.

scope of this review (see reviews *Nicoll, 2017*; *Volianskis et al., 2015*). Instead, we focus on variability in LTP due to stimulation at different sites, frequencies, cellular locations, development, and aging.

Regional differences exist in LTP induction. Left CA3 input induces ipsilateral and contralateral LTP in CA1, whereas right CA3 stimulation does not evoke LTP. CA1 synapses receiving left CA3 input show smaller spines with increased expression of the NMDA receptor (NMDAR) subunit GluN2B (*Kohl et al., 2011*; *Shinohara et al., 2008*), which evidence suggests is more effective than GluN2A in mediating LTP (*Shipton & Paulsen, 2014*; *Yashiro & Philpot, 2008*; *El-Gaby, Shipton & Paulsen, 2014*; *Sobczyk, Scheuss & Svoboda, 2005*). The relative expression of GluN2B to GluN2A subunits decreases during development (*Yashiro & Philpot, 2008*), but this change may be negligible at left CA3-CA1 synapses, causing them to remain in an "immature plastic state" (*El-Gaby, Shipton & Paulsen, 2014*). NMDAR expression is decreased in the ventral (temporal) compared to dorsal (septal) CA1 (*Martens & Wree, 2001*). LTP is correspondingly less in ventral

relative to dorsal PCs (*Maggio & Segal, 2007*; *Maruki et al., 2001*; *Papatheodoropoulos & Kostopoulos, 2000a*; *Milior et al., 2016*). Around 60% of dorsal PCs show robust LTP, while 57% of ventral PCs show no LTP (*Papatheodoropoulos & Kostopoulos, 2000a*). Ventral PCs that do show LTP show smaller changes in EPSP responses compared to dorsal PCs (*Maggio & Segal, 2007*; *Milior et al., 2016*). Interestingly, the cells that are less plastic are also more excitable. Ventral PCs have depolarized resting membrane potentials, increased input resistance, and fire at lower current stimulation amplitudes relative to dorsal PCs (*Malik et al., 2015*; *Milior et al., 2016*). (For background information on the passive properties of neurons, like resting membrane potential and input resistance, see Ch. 6 in *Kandel et al., 2013*. Or, for more on these properties and their measurement specifically in CA1 PCS, see *Spruston & Johnston, 1992*; *Staff et al., 2000*; *Zemankovics et al., 2010*). PCs that undergo successful LTP show decreased excitability, with lower input resistance and larger current amplitudes needed to evoke spiking, compared to prior to LTP. These changes result from a LTP-induced, NMDA-dependent increase in the hyperpolarization-activated cation current, $I_H$ (*Fan et al., 2005*; *Narayanan & Johnston, 2007*) (for reviews on H channels, see *Robinson & Siegelbaum, 2003*; *He et al., 2014*). The H current appears to act as a 'brake' on NMDA-dependent responses (*Otmakhova & Lisman, 2004*), and its involvement in LTP is pathway specific. Deletion of the H channel-encoding gene, HCN1, in mice increases LTP via PP but not SC inputs (*Nolan et al., 2004*).

Another form of LTP depends not on NMDARs but on voltage-dependent $Ca^{2+}$ channels (VDCCs) (*Grover & Teyler, 1990*; *Cavus & Teyler, 1996*; *Morgan & Teyler, 1999*). In CA1, depolarization of the dendrites via synaptic stimulation and/or bAPs can activate high-threshold VDCCs, leading to the generation of dendritic $Ca^{2+}$ spikes—large, regenerative depolarizations that are slower and longer-lasting than typical $Na^+$-based spikes (*Kamondi, Acsády & Buzsáki, 1998*; *Golding et al., 1999*) (for review see *Manita et al., 2017*). $Ca^{2+}$ entry during dendritic spikes is important for LTP induction (*Golding, Staff & Spruston, 2002*). Blocking VDCCs, especially L-type $Ca^{2+}$ channels, reduces LTP (*Cavus & Teyler, 1996*; *Freir & Herron, 2003*; *Morgan & Teyler, 1999*; *Golding, Staff & Spruston, 2002*). Though VDCC-dependent LTP is found in both young and old animals, the relative importance of this type of plasticity seems to change with aging. NMDA-dependent LTP is reduced in aged animals (*Shankar, Teyler & Robbins, 1998*; *Boric et al., 2008*; *Robillard et al., 2011*), while VDCC-dependent LTP is increased, compared to young controls (*Shankar, Teyler & Robbins, 1998*; *Robillard et al., 2011*), particularly in aged rats that show successful learning (*Boric et al., 2008*). When both forms are functional, LTP is equivalent in young and aged animals, demonstrating the compensatory role played by VDCCs. This could explain why some studies find age-related deficits in LTP, while others do not. Some stimulation protocols activate only NMDA-dependent LTP, revealing a deficit in aged animals, while other protocols activate VDCC-dependent LTP, allowing compensation (*Shankar, Teyler & Robbins, 1998*) (for review see *Rosenzweig & Barnes, 2003*).

LTP is often induced using high-frequency or tetanic stimulation (HFS). In recent years, theta burst stimulation (TBS) has been used as an arguably more realistic reproduction of endogenous hippocampal activity (for review see *Larson & Munkácsy, 2015*). Studies report TBS is more effective than HFS in inducing LTP (*Larson & Munkácsy, 2015*), especially

if bursts fall on the positive phase of endogenous theta rhythms (see 'Theta rhythm and phase precession'). If bursts fall instead on the negative phase, TBS induces depotentiation (*Hölscher, Anwyl & Rowan, 1997*) or long-term depression (LTD) (*Hyman et al., 2003*). In some studies, young and aged animals show equivalent LTP induced by HFS (*Moore, Browning & Rose, 1993*), while aged rats show reduced-magnitude LTP induced by TBS (*Moore, Browning & Rose, 1993*; *Rex et al., 2005*). In other studies, HFS LTP is reduced in aged animals but can recover to normal levels after chelation of excess zinc in aged cells (*Shetty, Sharma & Sajikumar, 2017*). Learning-impaired aged rats show decreased TBS-induced LTP, but no differences in LTP induced by HFS (*Tombaugh et al., 2002*). Aged rats show equivalent potentiation of dendritic EPSPs but weaker potentiation of somatic population spikes compared to young animals (*Deupree, Bradley & Turner, 1993*). *Deupree, Bradley & Turner (1993)* speculate this indicates functional plasticity of individual synapses but deficient dendritic integration and somatic signaling in aged animals.

LTP can also occur following pairing of low-frequency presynaptic stimulation with postsynaptic depolarization (*Campanac & Debanne, 2008*; *Andrade-Talavera et al., 2016*). LTP resulting from paired stimulation is one type of spike timing-dependent plasticity (STDP) in which the temporal order of spiking determines the type of plasticity induced; if presynaptic spiking occurs prior and close in time to postsynaptic spiking, then LTP occurs, while the reverse order (post →pre) produces long-term depression (LTD) (*Bi & Poo, 1998*; *Campanac & Debanne, 2008*; *Andrade-Talavera et al., 2016*) (for reviews on STDP see *Edelmann, Cepeda-Prado & Leßmann, 2017*; *Caporale & Dan, 2008*; *Dan & Poo, 2004*). Postsynaptic stimulation can come in the form of a single spike or a burst of APs, and the relative ability of these two stimuli to induce LTP changes with age. While single-spike stimulation can induce paired LTP in juvenile (P9-14) mice, a postsynaptic burst is required to generate the same level of potentiation in young adult (P22-28) mice (*Meredith, Floyer-Lea & Paulsen, 2003*). The same results are observed when comparing juvenile (P12-P15) to adult (P25-43) rats, and may be related to developmental increases in GABAergic inhibition (*Meredith, Floyer-Lea & Paulsen, 2003*).

## SPONTANEOUS AND NETWORK ACTIVITY

To understand the complete electrophysiological repertoire of neurons, we must see how they respond spontaneously under different behavioral conditions.

### Place field firing

Some CA1 PCs respond as an animal traverses specific areas in an environment (*O'Keefe & Dostrovsky, 1971*; *O'Keefe & Conway, 1978*; *Wlson & McNaughton, 1993*; *Best, White & Minai, 2001*). These place cells increase their firing rate in defined place fields, which develop when an animal explores either a physical or virtual space (*Harvey et al., 2009*). Some studies give the impression that spatial coding is the only function of CA1 PCs. However, Wiener cautions we should not define these cells only by their place-related firing since their activity is "highly plastic" and context-dependent (*Wiener, 1996*).

Not all PCs develop place fields. Between 30% and 70% of PCs whose activity is evoked by stimulation, or by select behavioral states, do not show spontaneous activity or

place-related firing in a maze environment (*Thompson & Best, 1989*; *Wlson & McNaughton, 1993*; *Harvey et al., 2009*; *Lee et al., 2004*). Spontaneously active place cells have distinct biophysical properties from so-called 'silent' cells, including increased excitability and plateau depolarizations (*Lee, Lin & Lee, 2012*). Depolarizing silent cells with somatic current injection during exploration induces the appearance of place fields (*Lee, Lin & Lee, 2012*). Thus, these cells receive spatial input but are unable to respond without a 'boost'. Around 20% of silent cells spontaneously and abruptly convert into place cells, which correlates with increased interneuron input (*Frank, Brown & Stanley, 2006*).

Most PCs have place-related firing in only one environment, but decrease their firing or go silent in others (*Thompson & Best, 1989*; *Ziv et al., 2013*). Just 1–3% of PCs have place-related activity in all environments tested (*Thompson & Best, 1989*; *Ziv et al., 2013*). PCs often have a single place field in small environments, but show multiple place fields when exploring a larger enclosure (*Park, Dvorak & Fenton, 2011*) (see Fig. 4). While overall representation of an environment is relatively stable, with 30% of PCs showing place firing, individual PCs drop in or out of the group of active cells, showing only a 15–25% overlap between recording sessions and demonstrating a "a day-to-day dynamism at the cellular level" (*Ziv et al., 2013*). PCs can drop in or out of the spatial representation in response to rotation of environmental cues (*Lee et al., 2004*). Many PCs increase their firing rate during place field traversal ('on' response), but some decrease their firing ('off' response) (*Thompson & Best, 1989*). 'On' and 'off' responses in a single PC are occasionally seen in the same environment.

PCs in different hippocampal areas show distinct spatial coding. While dorsal and ventral PCs show similar firing during resting and sleep, a larger percentage of dorsal than ventral cells show place fields during exploration (*Jung, Wiener & McNaughton, 1994*). Dorsal PCs show high spatial specificity. In contrast, ventral place fields are so much larger that some researchers suggest calling these "context" instead of "place" cells (*Maurer et al., 2005*). PCs in the middle hippocampal region have place fields intermediate in size, half that of ventral fields but larger than dorsal ones (*Maurer et al., 2005*). PCs located closer to the stratum oriens (called deep PCs) are more likely to form place fields than PCs closer to the stratum radiatum (superficial PCs) (*Mizuseki et al., 2011*; *Danielson et al., 2016*). Deep PCs tend to form place fields which are more plastic, while superficial PCs have place fields which are more stable over time (*Danielson et al., 2016*). Deeps PCs are also more attuned to specific landmarks, while superficial PCs respond more to context, i.e., the overall arrangement of cues in a spatial environment (*Geiller et al., 2017*). Ionic currents and their effect on PC firing rates can affect place field size and stability. For example, in dorsal PCs, deletion of the H channel-encoding gene, HCN1, results in larger but more stable place fields (*Hussaini et al., 2011*).

Place-related firing can change with age and experience. Pre-weaning rats (<P22) show an increased tendency to form place fields at boundaries and poorer spatial accuracy in other areas of an environment, compared to post-weaning and adult rats (*Muessig et al., 2015*) (Fig. 4). Place fields increase in size with increasing exploration (*Mehta, Barnes & McNaughton, 1997*; *Mehta, Quirk & Wilson, 2000*; *Huxter, Miranda & Dias, 2012*). Place field expansion occurs on the first day of multiple runs through an environment, but not on

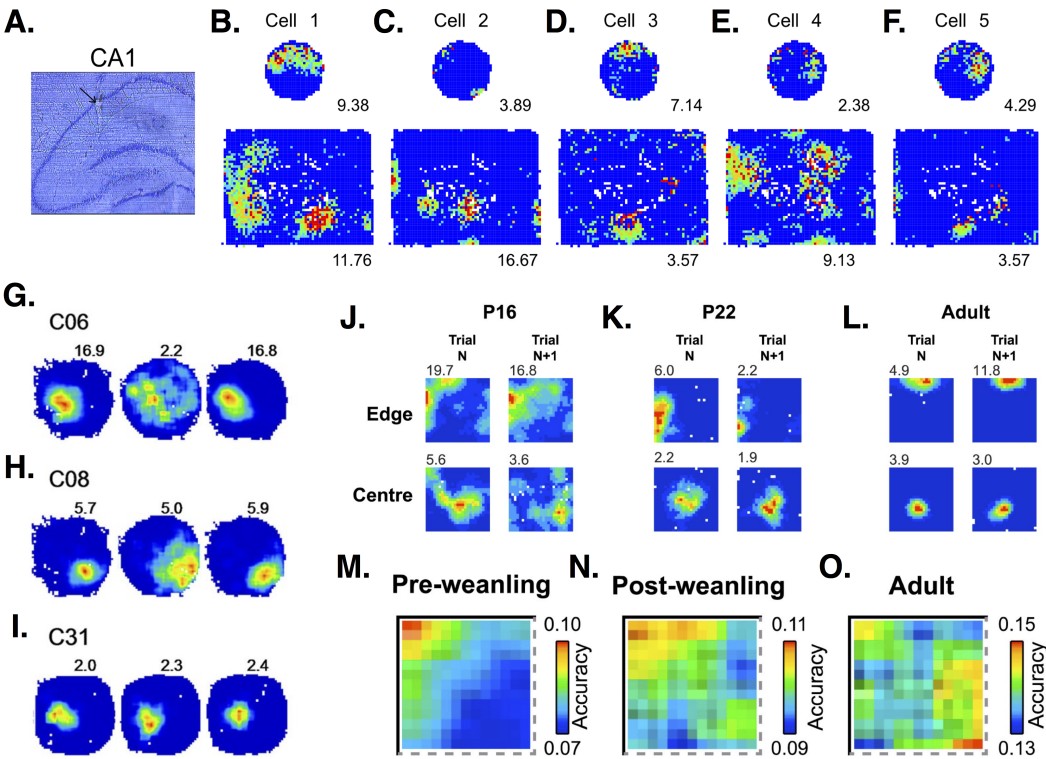

**Figure 4** **Place cell firing in CA1 PCs.** (A) Histological section showing the place cell recording site within CA1 region of the hippocampus. (B–F) Firing rate maps show that CA1 PCs in rats tend to develop single place fields in a small, circular environment but multiple place fields in a larger, square box enclosure. Maps from 5 different cells show some of the variability in spatial representations across PCs. Images (A–F) are from *Park, Dvorak & Fenton (2011)*. (G–I) Firing rate maps in three different CA1 PCs show variable responses to light (left-hand circle), dark (middle circle), light (right-hand circle) conditions. Cell C31 (I) shows virtually no change in firing, while cells C06 (G) and C08 (H) show decreased frequencies and less-defined fields under dark conditions. Images (G–I) are from *Zhang et al. (2014)*. (J–L) Firing rate maps at an initial time point (trial N) and 15 min later (trial $N + 1$) in postnatal day 16 (P16), P22, and adult rats. Developing rats have more diffuse and less stable place fields than adults. Images (J–L) are from *Muessig et al. (2015)*. (M–O) Place cells in pre-weanling rats (P14-21) have good spatial accuracy at specific boundary points, but poor accuracy in the rest of the environment. In contrast, cells in more developed (post-weanling) or adult rats have good accuracy throughout the environment. Images (M–O) are also from *Muessig et al. (2015)*. All figures reused under the CC-BY license.

subsequent days (*Lee, Rao & Knierim, 2004*). Firing rates within place fields also increase and the center of mass of the firing rate distribution shifts backwards in space (*Mehta, Barnes & McNaughton, 1997*; *Mehta, Quirk & Wilson, 2000*; *Lee, Rao & Knierim, 2004*; *Huxter, Miranda & Dias, 2012*). Place field shape also changes with experience. *Mehta, Quirk & Wilson (2000)* report that 78% of place fields are asymmetric and negatively skewed, with firing rates ∼35% higher in the second versus the first half of activity. Place field asymmetry is not present when animals are first introduced into an environment, but develops with increasing exploration. This progression from symmetric to asymmetric place fields occurs daily, even in previously explored environments (*Mehta, Quirk & Wilson, 2000*). *Huxter, Miranda & Dias (2012)* also report changes in place field shape, but

in contrast to Mehta *et al.* they find that place fields are initially positively skewed and become symmetrical with increasing experience. Experience-related changes in place cell firing are impaired in old animals. Aged rats have smaller place fields than young rats, and show little to no expansion with experience (*Shen et al., 1997*). Increases in firing rate with experience are also smaller in aged animals (*Shen et al., 1997*).

## Theta rhythm and phase precession

Rhythmic oscillations in theta frequency (6–12 Hz) are recorded at cellular and population levels in CA1 (*Buzsáki, 2002*; *Hasselmo, 2005*). While several studies demonstrate the importance of CA3 input in theta rhythm generation, other studies show that CA1 is capable of sustaining an intrinsic rhythm when isolated (*Goutagny, Jackson & Williams, 2009*). Dendritic depolarization in CA1 PCs occurs during the positive phase (peak) of extracellularly-recorded theta (*Kamondi et al., 1998*). In contrast, somatic depolarization occurs primarily during the negative phase (trough) of theta (*Kamondi et al., 1998*). The temporal relationship of PC firing to theta is variable and depends on cellular properties and behavior. Buzsáki and colleagues report that while most PCs fire on the negative phase (*Csicsvari et al., 1999*; *Dragoi & Buzsáki, 2006*; *Kamondi et al., 1998*; *Mizuseki et al., 2011*; *Ylinen et al., 1995*), a small percentage of weakly-activated cells fire on the positive phase of theta (*Buzsáki, 2002*; *Csicsvari et al., 1999*). They also report that a large percentage of PCs are slow-spiking or silent during theta and argue this could bias analysis of PC activity (*Csicsvari et al., 1998*; *Buzsáki, 2002*; *Harris et al., 2000*).

PC theta firing preference depends on cell location and behavioral state (*Mizuseki et al., 2011*). During exploratory behavior, PCs throughout the CA1 sublayers tend to phase-lock their firing with the trough of theta. During REM sleep, however, while stratum radiatum neurons fire around the trough, ~50% of stratum oriens neurons fire instead during the peak. CA1 PCs in distinct sublayers show other biophysical differences, leading *Mizuseki et al. (2011)* to classify them as subpopulations.

PC theta firing preference can shift during exploratory behavior. *O'Keefe & Recce (1993)* report that as rats traverse a place field, CA1 PCs fire progressively earlier with respect to extracellular theta. This phenomenon, referred to as phase precession, has been studied extensively and established as a characteristic feature of place cell activity (*Kamondi et al., 1998*; *Skaggs et al., 1996*; *Mehta, Barnes & McNaughton, 1997*). The extent of phase precession depends on experience. At the population level, PCs show a 2-fold increase in the correlation between phase and spatial location after multiple trial runs through an environment (*Mehta, Lee & Wilson, 2002*). There is a dissociation between the emergence of theta-related firing at the single cell versus population level. While phase precession is present in single CA1 PCs from the first trial run through a novel environment, coordination of population-level neural activity with respect to theta is not recorded until the second trial (*Feng, Silva & Foster, 2015*).

Stimulation during theta activity results in different types of plasticity depending on when it is delivered. *Huerta & Lisman (1995)* report that a single burst delivered at the peak of extracellular theta results in LTP, while the same stimulation delivered near the trough leads to depotentiation. Like Huerta and Lisman, *Hölscher, Anwyl & Rowan (1997)*

found that stimulation on the positive phase of theta leads to LTP, while stimulation on the negative phase depotentiates, but does not depress, synaptic responses. Stimulation on the zero phase of theta does not change synaptic responses. In contrast, Hyman and colleagues *(2003)* found the same results for positive phase stimulation, but report depression, not just depotentiation, after negative-phase stimulation. They argue their results may differ due to the temporal precision of their stimulation, hitting exactly at the trough rather than elsewhere on the negative phase.

Theta activity is altered, albeit in small ways, during aging. Middle-aged rats show lower theta amplitude relative to young rats during waking and at multiple running speeds (*Huxter, Miranda & Dias, 2012*; *Kuo et al., 2010*). Theta frequency is higher in middle-aged rats during running, though this change may be specific to early aging (*Huxter, Miranda & Dias, 2012*), as older animals show decreased theta frequency (*Cooper, Prinz & Marsh, 1975*; *Markowska, Olton & Givens, 1995*; *Shen et al., 1997*; *Jacobson et al., 2013*). In young rats, theta frequency (*Shen et al., 1997*) and amplitude (*Huxter, Miranda & Dias, 2012*) increase at higher running speeds. These speed-related increases are smaller in aged animals. Likewise, increases in theta power seen in young animals as they transition from resting to active states (*Jacobson et al., 2013*) or due to exercise (*Kuo et al., 2010*) are smaller in aging animals. Novel environments induce smaller increases in theta power for aged versus in young rats (*Jacobson et al., 2013*). In contrast to differences seen during activity, there is no difference in theta recorded from young or aged animals during REM sleep (*Shen et al., 1997*). Theta phase precession is also not different in aging animals (*Huxter, Miranda & Dias, 2012*; *Shen et al., 1997*).

## Gamma oscillations

CA1 displays spontaneous oscillations in the gamma frequency (∼30–150 Hz)[1] (*Buzsáki & Wang, 2012*; *Colgin & Moser, 2010*; *Leung, 1998*; *Lisman, 2005*). Gamma oscillations are nested within theta cycles, and larger during theta-related activities, such as exploratory behaviors and REM sleep (*Bragin et al., 1995*; *Buzsáki, Leung & Vanderwolf, 1983*; *Csicsvari et al., 2003*). Changes in the frequency (*Bragin et al., 1995*) and amplitude (*Bragin et al., 1995*; *Jacobson et al., 2013*; *Penttonen et al., 1998*) of gamma oscillations correlate with changes in corresponding theta measures.

Gamma oscillations can be divided into three distinct bands (*Belluscio et al., 2012*; *Schomburg et al., 2014*): (1) slow gamma (∼30–50 Hz), (2) mid gamma (∼50–90 Hz), and (3) fast gamma (90–150 Hz), sometimes called the epsilon band (*Belluscio et al., 2012*; *Buzsáki & Wang, 2012*; *Schomburg et al., 2014*). Each frequency band shows a different theta phase preference, though reports vary. *Belluscio et al. (2012)* report that slow gamma power is largest on the descending phase of theta, mid gamma largest near the peak, and fast gamma largest near the trough (*Schomburg et al., 2014*). *Scheffer-Teixeira et al. (2012)* find that fast gamma is instead largest on the descending phase of theta. *Colgin et al. (2009)* who collapse mid and fast gamma into one band (65–140 Hz), report that the largest power is near the trough of theta. Theta phase preference can change depending on behavioral state. The peak power of fast gamma shifts from theta trough to peak during exploratory behavior versus REM sleep, respectively (*Belluscio et al., 2012*).

[1] Gamma range is typically ∼30–80 Hz, but upper and lower limits vary depending on the reference.

ML... 

CA1 gamma oscillations vary with depth. Slow gamma is largest in the stratum (s.) radiatum, while mid gamma is largest in the s. oriens/pyramidale and s. lacunosum-moleculare (*Belluscio et al., 2012*). Fast gamma dominates a small section of the s. pyramidale. While slow gamma power in the s. radiatum decreases during REM sleep, mid gamma power in the s. lacunosum-moleculare increases (*Schomburg et al., 2014*). Gamma-theta coupling also varies by layer, but results differ across studies. *Scheffer-Teixeira et al. (2012)* report that mid gamma-theta coupling is strongest in the s. lacunosum-moleculare, but fast gamma-theta coupling is strongest in s. oriens-alveus. They find no coupling between slow gamma and theta rhythms. In contrast, *Schomburg et al. (2014)* report slow gamma-theta coupling exists and is strongest in the s. radiatum. Gamma frequency bands and their layer preferences relate to different CA1 inputs. Slow gamma in CA1 correlates with oscillations recorded from CA3, while mid to fast gamma correlates with oscillations in the entorhinal cortex (*Colgin et al., 2009*; *Schomburg et al., 2014*).

The percentage of CA1 PCs phase-locking their firing to gamma varies with behavioral state, recording distance, and gamma frequency. *Senior et al. (2008)* report that during waking ∼32% of PCs phase lock their firing to 30–80 Hz gamma, versus only 4% during REM sleep. *Mizuseki et al. (2011)* report that ∼27% of PCs gamma phase lock during maze runs, but only 10% do so during REM sleep. *Csicsvari et al. (2003)* report that ∼43% of PCs phase lock to locally recorded 30–80 Hz oscillations but only 13% to distal fields. Show that 18% of PCs phase lock to slow gamma, 36% to mid-range gamma, and 75% to fast gamma (*Belluscio et al., 2012*). *Schomburg et al. (2014)* also report that CA1 PCs preferentially phase lock to fast gamma.

CA1 PCs vary with respect to gamma phase preference. During exploratory activity, some PCs fire during the rising phase of gamma, while others fire at the trough (*Belluscio et al., 2012*; *Mizuseki et al., 2011*; *Senior et al., 2008*). PCs firing on the rising phase during active states shift their preference during REM sleep, while the majority of the population fires at the trough (*Mizuseki et al., 2011*; *Senior et al., 2008*). Rising-phase versus trough-phase PCs are also different with respect to firing rate, interspike interval, bursting propensity, action potential shape, magnitude of afterpotentials, theta phase preference, and activity during theta phase precession (*Mizuseki et al., 2011*; *Senior et al., 2008*). Rising-phase PCs are most likely to be found close to the s. oriens, while trough-phase PCs are located closer to the s. radiatum (*Mizuseki et al., 2011*).

Gamma-related activity can vary with experience and in a task-dependent manner. PCs preferentially phase lock their firing to either familiar or novel environments (*Senior et al., 2008*). Relative gamma power increases after learning, but only on certain types of task (*Muzzio et al., 2009*). For some tasks but not others, PCs increasingly phase lock their firing to gamma oscillations, particularly those occurring seconds before reward delivery (*Muzzio et al., 2009*). Subtle effects of aging on gamma oscillations have been reported. In middle-aged rats, the increase in normalized gamma amplitude peaks at larger running speeds than in younger animals (*Huxter, Miranda & Dias, 2012*). In old rats, gamma amplitude changes little during the theta cycle, in contrast to the stronger modulation seen in young rats (*Jacobson et al., 2013*). The increase in gamma power seen when young rats transition from resting to running occurs but is less prominent in old rats

(*Jacobson et al., 2013*). Gamma coherence between adjacent recording sites is lower in aged versus young rats, both within and across CA1 layers (*Jacobson et al., 2015*).

## DISCUSSION

### Text and data mining to explore cellular heterogeneity

There is a wealth of information on cellular heterogeneity in the existing literature. However, finding all the relevant articles, extracting key information, and comparing results across multiple studies is a task not easily or efficiently done by hand. With text and data mining (TDM), researchers are now automating these tasks to perform large-scale meta-analyses (*Rebholz-Schuhmann, Oellrich & Hoehndorf, 2012*). A recent study by *Tripathy et al. (2015)* used TDM to examine variation in biophysical properties, such as resting membrane potential and input resistance, across multiple neuron types, including CA1 PCs. Much of the variation they observed could be explained by methodological differences, while significant variation was unexplained and could be due to cellular heterogeneity. Cluster analysis of six biophysical measures confirmed known neuron classes, but also revealed new classes based on previously unidentified similarities between cells (*Tripathy et al., 2015*). *Wheeler et al. (2015)* mined the hippocampal literature and identified 122 neuron types based on biochemical, electrophysiological, and morphological features. These studies demonstrate the power of TDM approaches and suggest there are more discoveries to be made by mining the neuroscience literature (*Akil, Martone & Van Essen, 2011*).

We propose mining the literature for phrases related to different aspects of electrical activity, such as "delayed firing" or "spike latency". Automated download of all figures with electrical traces from articles including these phrases will facilitate large-scale comparisons within and across neuron types. Mining can be expanded beyond just the text to data within figures in published articles, made possible with tools developed by projects like Content Mine (http://contentmine.org/) (*Hartgerink & Murray-Rust, 2017*). Electrical traces can be extracted from figures and transformed into raw time and voltage data to perform new analyses not done in the original work.

A limitation to this approach is the possible bias present in published articles. Researchers select a few examples from their electrophysiological data to include in the final paper. These recordings may be representative of the majority of recordings, but may exclude some of the diversity found in the sample. In extreme cases, researchers may even remove recordings with different firing patterns from their analysis, assuming they are outliers or possibly different cell types. Proper large-scale meta-analyses of firing pattern diversity will require access to complete, original data sets, confirming the importance of data sharing initiatives for TDM research (*Molloy, 2011*; *Mons et al., 2011*). Copyright and article licensing reform is also crucial for TDM research, as certain laws restrict the mining of data from the published literature (*Haeussler et al., 2015*; *Handke, Guibault & Vallbé, 2015*; *Triaille, de Meeûs d'Argenteuil & de Francquen, 2014*; *Reichman & Okediji, 2012*).

### Dynamical systems theory to study firing patterns

Neurons can be thought of as families of dynamical systems (*Herrera-Valdez et al., 2013*), which evolve over time according to specific rules (*Ermentrout & Terman, 2010*; *Izhikevich,*

*2007*; *Rubin & Rinzel, 2016*; *Terman, 2005*). Many of the things we look for experimentally when studying neuron responsiveness have their equivalent in dynamical systems. For example, we measure the resting membrane potential (RMP) of a neuron (if it exists) to find out where the cell 'sits' if unstimulated. We inject a small amount of current into the neuron to observe how this changes the membrane potential, and how long the neuron takes to return to rest once stimulation is removed. Sometimes we see that the neuron returns smoothly to rest, while in other cases, the neuron displays subthreshold oscillations of diminishing amplitude before settling again at the RMP (*Izhikevich, 2007*).

In dynamical systems, the RMP is a fixed point (FP) where the variables of the system are not changing (*Izhikevich, 2007*). In particular, the RMP is a special type of FP called an attractor, which means the system tends to return to this point after a small perturbation (e.g., a small current injection). We can learn something about the computational properties of a neuron by studying the local dynamics around different types of FPs. If the system is near one type of FP attractor, called a node, the system will return to this point after a small perturbation smoothly and without oscillating. In contrast, if the system is near a different type of FP attractor, called a focus, it will return to this point after a small perturbation in a spiral trajectory, causing subthreshold oscillations of diminishing amplitude. In neurons, this is important because it means the system can resonate with different frequency inputs (*Izhikevich, 2007*).

Experimentalists are interested in the transitions neurons make as they go from resting to spiking (*Izhikevich, 2007*). A common protocol involves injecting progressively larger current steps to see how a neuron responds. We start off with a small magnitude current that is insufficient to produce spiking. Next, a slightly larger current injection may produce small membrane potential oscillations during the stimulation, but again, no spikes. If we inject a still larger amount of current, then the neuron may fire a single action potential, which is characterized by a rapid and large increase in the membrane potential. The neuron then repolarizes, either fully, or partially to remain steadily at a depolarized potential until the current injection ceases. Finally, if we inject even more current, many neurons will respond by firing repetitively, and sustain this activity until the stimulation ends. It is clear from this example that there is a point at which increasing the stimulation amplitude induces a qualitative change in the behavior of the neuron. The protocol allows the researcher to observe the neuron as it moves from a resting to a repetitive spiking state, and furthermore, to see the effect of changing the stimulation amplitude on this transition (*Izhikevich, 2007*).

In dynamical systems, we are likewise interested in transitions between different states (e.g., rest and repetitive spiking), and the effect that varying parameters (e.g., the stimulus amplitude) has on those transitions (*Izhikevich, 2007*; *Rubin & Rinzel, 2016*; *Terman, 2005*; *Ermentrout & Terman, 2010*). Transitions that produce qualitative changes in the behavior of a system are called bifurcations. Studying the bifurcations that neurons undergo when going from rest to spiking states can reveal the types of firing patterns a neuron is capable of producing. Furthermore, while neurons may appear to have many different firing patterns, there are only a small number of bifurcations that produce rest-to-spiking transitions. We can use this knowledge to study whether neurons share a common firing mechanism,

and potentially reduce the number of designated cell classes by grouping neurons accordingly. We may also be able to predict when a cell might shift between different firing mechanisms due to changes in parameters (*Izhikevich, 2007*; *Rubin & Rinzel, 2016*; *Ermentrout & Terman, 2010*; *Terman, 2005*), such as the density or kinetics of channels in the membrane (*Herrera-Valdez et al., 2013*; *Prescott, De Koninck & Sejnowski, 2008*).

In 1948, Hodgkin grouped neurons into classes based on their firing in response to a range of current stimulation amplitudes (*Hodgkin, 1948*). In 1989, Rinzel and Ermentrout showed that these three firing classes were related to distinct bifurcations (*Rinzel & Ermentrout, 1998*). Subsequent studies have reported that model neurons can switch between Hodgkin's three classes, and that their transitions from rest to spiking depend on the balance and timing of specific ion currents (*Prescott, De Koninck & Sejnowski, 2008*). Previous work by one of the present authors (ECM) has shown that the type of bifurcation producing the transition into spiking when the current stimulation amplitude is changing, and correspondingly the presence of short or long delays to first spike in a model motor neuron, depends on the relative expression of $Na^+$ and $K^+$ channels (*Herrera-Valdez et al., 2013*). In sum, bifurcation analysis can demonstrate under what cellular conditions certain firing patterns may emerge, as well as show us the functionally equivalent combinations of channels that produce the same rest-to-spiking transitions. Such studies could help reconcile some of the seemingly contradictory experimental results seen in CA1 PCs, for example, with respect to the currents responsible for generating AHPs and other electrical behaviors.

## Mathematical modeling to study effects of cellular heterogeneity on network function

While differences in firing patterns may be subtle, or the percentage of PCs deviating from what are considered classical firing behaviors may be small, variability could have important effects on network function (*Padmanabhan & Urban, 2010*). Unfortunately, it is difficult to properly test the effects of cellular heterogeneity in intact networks. First, characterizing the full range of biophysical properties displayed by cells within a network is often not possible. Second, it is difficult to know the percentage of cells in a network demonstrating different biophysical properties, thus limiting our ability to determine the number of critical cells required to see an effect. Third, manipulating cells with a given property is difficult without also affecting other members of the network.

Mathematical modeling allows the targeted perturbation of select biophysical properties in single neurons. Since there is no homeostatic compensation, as often observed in real neural networks (*Marder & Goaillard, 2006*), simulations can be compared to experimental results to determine whether a change in a given parameter (e.g., the expression of a certain ion channel) is sufficient to produce specific firing patterns. While historically most models have assumed neurons within a population to be relatively homogeneous, heterogeneity can be made a model feature to explore effects (*Golomb & Rinzel, 1993*; *Pinsky & Rinzel, 1994*; *Mejias & Longtin, 2012*; *Ferguson et al., 2015*). Model networks can be assembled with a known percentage of cells displaying a given biophysical property to determine the critical mass necessary to produce a given output. Synaptic partners and the strength

of their connections can be varied to explore how cellular heterogeneity interacts with network organization to produce circuit output, or compensate in cases of altered cellular excitability. In particular, PCs can be modeled to represent cells in juvenile, adult, or aged animals to explore how changes in biophysical properties during developmental stages may affect circuit function.

To build a network model that will allow us to investigate these issues, we will first need to build the component cells using models that are both biophysical and relatively simple. Several models exist to investigate electrical activity in CA1 PCs (*Bianchi et al., 2012*; *Golomb, Yue & Yaari, 2006*; *Gu et al., 2005*; *Poirazi, Brannon & Mel, 2003*; *Shah et al., 2008*; *Shao et al., 1999*; *Nowacki et al., 2011*). Many attempt to represent PC morphology using multiple compartments (*Gu et al., 2005*; *Poirazi, Brannon & Mel, 2003*; *Shah et al., 2008*; *Shao et al., 1999*). A few single-compartment models of CA1 PCs exist, but they have up to 10 ionic currents and as many as 17 variables (*Bianchi et al., 2012*; *Golomb, Yue & Yaari, 2006*; *Nowacki et al., 2011*). As the number of variables and compartments increases, mathematical analysis becomes harder and limits our ability to understand the influence of certain parameters, as well as the possibility of constructing simple network models, since the computational load is high. There are simpler models of CA1 PCs, but which do not include ionic currents (*Ferguson et al., 2014*), and are therefore not suited for the questions we have posed here.

From the literature discussed herein, it is clear that a minimal biophysical model of a CA1 PC will need to include at least three dimensions to be able to reproduce repetitive firing with spike frequency adaptation, conditional bursting, and endogenous bursting. The model should contain transient $Na^+$ and delayed rectifier $K^+$ currents to produce basic spiking, and most likely $Ca^{2+}$ and $Ca^{2+}$-dependent currents to generate adaptation and bursting. To simulate aging, a first place to start could be varying $Ca^{2+}$ channel expression and handling (*Oh et al., 2013*). Stimulation protocols used experimentally to induce short- and long-term potentiation can be simulated to systematically test the effects of cellular aging on plasticity. Using a minimal biophysical model to construct the single cells will aid in subsequent network studies, as the components will be as computationally inexpensive as possible while still reproducing a range of cellular behaviors and allowing us to ask questions about the role of different channels.

## CONCLUSIONS

CA1 pyramidal cells demonstrate an incredible diversity and plasticity in their responses to somatic current injection, synaptic stimulation, and spontaneous network-related activity. Based on these data, it is clear that the functional heterogeneity of PCs (and likely most neuronal populations) is far more significant than is often appreciated. In some cases, cellular subpopulations may be stable (*Graves et al., 2012*), while in other cases, subpopulations may be fluid, with cells transitioning between different activity profiles depending on development, experience, and aging, among other factors. Further studies, both experimental and computational, are needed to explore the breadth of

PC heterogeneity and understand its impact on hippocampal circuitry. In particular, to make significant advances, we believe it is necessary to increase interactions between experimental and theoretical researchers, and design studies that integrate their different but complementary approaches (*Skinner, 2013*; *Shou et al., 2015*).

## ACKNOWLEDGEMENTS

The authors thank Marco A. Herrera Valdez and Alberto Granato for helpful comments that improved this manuscript.

### Funding

This work was supported in part by the Natural Sciences and Engineering Research Council of Canada, as well as the Ontario Mental Health Foundation. This work was also supported by the Dirección General de Asuntos del Personal Académico, Programa de Apoyo a Proyectos de Investigación e Innovación Tecnológica at the Universidad Nacional Autónoma de México (UNAM-DGAPA-PAPIIT IA209817). There was no additional external funding received for this study. The funders had no role in study design, data collection and analysis, decision to publish, or preparation of the manuscript.

### Grant Disclosures

The following grant information was disclosed by the authors:
Natural Sciences and Engineering Research Council of Canada.
Ontario Mental Health Foundation.
Dirección General de Asuntos del Personal Académico, Programa de Apoyo a Proyectos de Investigación e Innovación Tecnológica at the Universidad Nacional Autónoma de México: UNAM-DGAPA-PAPIIT IA209817.

### Competing Interests

The authors declare there are no competing interests.

### Author Contributions

- Erin C. McKiernan wrote the paper, prepared figures and/or tables, reviewed drafts of the paper.
- Diano F. Marrone wrote the paper, reviewed drafts of the paper.

### Data Availability

This is a literature review paper. As such, the research involved in writing this article did not generate any data or code.

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
