# Peer review of "CA1 pyramidal cells have diverse biophysical properties, affected by development, experience, and aging"

_PeerJ, doi:10.7717/peerj.3836_

## Round 0.1 · original submission · Minor Revisions

As you can see, both reviewers were very positive about your manuscript, finding only minor points to address.

I advise you to deal with the points raised by reviewer 2.

·

Basic reporting

In this manuscript, the authors review the literature on the variability in the electrical activity of CA1 hippocampal pyramidal cells, considering responses to somatic current injection, synaptic stimulation, and spontaneous network-related activity. The layout is clear and the material is very well presented and discussed.

Experimental design

Being a review, the manuscript does not present any original material.

Validity of the findings

Being a review, the manuscript does not present any original findings.

Additional comments

I have no specific comments. In this manuscript, the authors review the literature on the variability in the electrical activity of CA1 hippocampal pyramidal cells, considering responses to somatic current injection, synaptic stimulation, and spontaneous network-related activity. The layout is clear and the material is very well presented and discussed.

·

Basic reporting

The review by McKiernan and Marrone deals with the electrophysiological properties of CA1 pyramidal cells and with the changes occurring during development and aging.
This is a well-written, solid, very exhaustive review and can represent a starting point for future research in the field. Most of the relevant liteature is cited. The frequent reference to modeling studies is valuable. The discussion on data mining is original and interesting.

Experimental design

No comment

Validity of the findings

No comment

Additional comments

In my opinion, there are only a few minor points that the authors may wish to address before the publication of the paper.

Row 68: resting (subthreshold) properties such as RMP, time constant, input resistance, depolarizing sag are not discussed in a separate section, even though they are mentioned in the section regarding LTP (rows 359-361). A brief paragraph introducing these basic properties would be helpful (the authors may cite ref. 56 of the current version) As to the sag, it is worth mentioning the relevance of I(h) current for plasticity (Brager and Johnston, 2007, J Neurosci 27:13926-37) and pathology (e.g. Fan et al., 2008, Exp Neurol 212:415-21. doi: 10.1016/j.expneurol.2008.04.032).

The backpropagation of action potentials is never cited in the review (see, for instance, Golding et al., 2001, J Neurophysiol. 86:2998-3010).

The relationships among afterdepolarization, dendritic spikes, and bursting are addressed in several points throughout the manuscript (e.g. rows 238-239; rows 272-273). However, due to the relevance of dendritic (calcium) spikes for plastic changes (pointed out at row 363) and other hippocampal functions, I guess that a few lines specifically devoted to dendritc spikes would be helpful (see, for instance, the section dealing with hippocampal dendritc spikes in Manita et al., 2017, Front Cell Neurosci 11:29. doi: 10.3389/fncel.2017.00029).

Section 4.3 (LTP): a brief outline of STDP in CA1 pyramidal neurons would be warranted (e.g., Campanac and Debanne, 2008, J Physiol 586:779-93).

---

## Round 0.2 · accepted · Accept

I have no additional comments

·

Basic reporting

This is a very exhaustive review and can represent a starting point for future research in the field. Most of the relevant liteature is cited. The frequent reference to modeling studies is valuable. The discussion on data mining is original and interesting.

Experimental design

Not applicable

Validity of the findings

Not applicable